# Chemical Composition and Bioactivity of Dill Seed (*Anethum graveolens* L.) Essential Oil from Plants Grown under Shading

**DOI:** 10.3390/plants13060886

**Published:** 2024-03-19

**Authors:** Lidija Milenković, Zoran S. Ilić, Ljiljana Stanojević, Bojana Danilović, Ljubomir Šunić, Žarko Kevrešan, Jelena Stanojević, Dragan Cvetković

**Affiliations:** 1Faculty of Agriculture, University of Priština in Kosovska Mitrovica, 38219 Lešak, Serbia; lidija.milenkovic@pr.ac.rs (L.M.); ljubomir.sunic@pr.ac.rs (L.Š.); 2Faculty of Technology, University of Niš, 16000 Leskovac, Serbia; ljilja_stanojevic@yahoo.com (L.S.); danilovic@tf.ni.ac.rs (B.D.); jelena_stanojevic@yahoo.com (J.S.); dragancvetkovic1977@yahoo.com (D.C.); 3Institute of Food Technology, University of Novi Sad, 21000 Novi Sad, Serbia; zarko.kevresan@fins.uns.ac.rs

**Keywords:** shading, dill seed, essential oil, content, chemical composition

## Abstract

This study determined the content and composition of dill seed (*Anethum graveolens* L.) essential oil under varying light conditions: non-shaded plants in open fields and plants covered with pearl shade nets (40% shade index). Essential oil was extracted using Clevenger hydrodistillation. The essential oil content was 4.63% for non-shaded plants and 4.81% for shaded plants. GC/MS analysis revealed twenty-one and twenty-two components in dill seed from non-shaded and shaded plants, respectively. The terpenic fraction of essential oil from non-shaded plants consisted mainly of oxygen-containing monoterpene derivatives (53.6%), with carvone (46.1%) as the primary component, followed by monoterpene hydrocarbons (46.4%), predominantly limonene (43.8%). Essential oil from shaded plants contained a higher content of carvone (49.8%) and a lower content of limonene (37.8%) compared to essential oil from non-shaded plants. Non-shaded plant essential oil exhibited stronger antioxidant activity (EC_50_ value: 26.04 mg mL^−1^) than shaded plant essential oil (54.23 mg mL^−1^). Dill seed essential oil showed the most potent antimicrobial activity (disc diffusion method) against *Escherichia coli* (inhibition zone: 15–18 mm). Shaded plants demonstrated a positive influence of essential oil against *Klebsiella pneumoniae*. Carvone and its derivatives, as the main components, hold significant potential in the food industry and alternative medicines. A practical implication of this study could be higher plant densities or intercropping of dill, as it thrives with minimal light.

## 1. Introduction

The majority of culinary herbs possess pleasant odors and the presence of antioxidants (polyphenolic compounds), making them significant for application in food industries [1] or drug development in the pharmaceutical and medical sector [2]. Dill (*Anethum graveolens* L.) is a leafy vegetable native to southern Europe and west Asia, and it grows as a wild plant in the Mediterranean region, where it was known to the ancient Greeks and Romans. In Serbia, it is cultivated in small areas as a spice vegetable in gardens and in the field for processing [3]. Dill, an annual species with a short growing season (vegetative phase: 90 days after sowing), requires approximately 210 days after sowing for seed production (fruiting stage) [4]. Its young, juicy twigs with needle-like leaves are used as a spice in dishes, fresh salads and processing vegetables. In Polish cuisine, fresh dill with specific flavor is the most widely used herbal spice [5]. Dill seeds, known for their aromatic, carminative and stimulant properties, are commonly used in food industries (whole or ground) for flavoring in meats, stews, pastries, bakery products and vinegars [6]. Additionally, they find applications in the cosmetic and pharmaceutical sectors, as well as in traditional medicine, being used to treat gastrointestinal problems, such as gas, indigestion, stomachache [7], to enhance milk production in nursing mothers, alleviate colic in babies [8], and they potentially act as a contraceptive agent [9]. Arab cultures employ dill seeds for various purposes, including as an appetizer, carminative, mouthwash, anthelmintic, antispasmodic and aphrodisiac [10]. Dill seed essential oil (DSEO) has been reported to possess antimicrobial [7,11], antifungal [12], antihyperlipidemic, diuretic, hypertensive, antiemetic and laxative effects [13].

Dill is increasingly recognized as an important culinary and medical plant and as a valuable source of bioactive compounds. Plants grown at low densities exhibit lower leaf mass but increased development of reproductive organs, particularly umbels with seed, compared to plants grown at high densities [14]. Dill should be grown as a sole crop to obtain the best seed yield; as an intercropping species in companion with onion and carrot plants; or as pre-crop, following crop, or cover crop in organic gardens. Dill seed essential oil (DSEO) demonstrates repellent and insecticidal activities [15], primarily attributed to its main component—carvone—which can be used during postharvest as a retardant to prevent potato sprouting [15].

Genetic factors (chemotype) and the interaction of environmental conditions (location) are pivotal in determining the content and composition of dill essential oil (DEO). The cultivar, developmental stage, plant part and harvest time play significant roles in producing essential oil (EO) from dill, which is an essential-oil-producing crop [16]. EO compounds accumulate and distribute differently across plant tissues and ontological phases. DEO extraction from different plant parts reveals its lowest content in leaves (0.08%), increasing to 1.10% in flowers and reaching its maximum (3.20%) in the full ripening stage with seeds [17]. The composition of DEO also depends on the plant part and developmental stage. The major EO components during the vegetative stage include α-phellandrene (46.33%), limonene (13.72%), β-phellandrene (11.71%) and p-cymene (17.88%), while p-cymene (33.42%), carvone (13.10%) and dill ether (19.63%) are detected in the flowering stage. Carvone and limonene have been reported as the main components of DEO [17,18]. The aroma of dill seeds is derived from carvone (spicy caraway) and limonene (fresh-citrus like). Monoterpenes (limonene and carvone) possess significant antibacterial activity due to lipophilic characters, allowing them to permeate into the membrane structure and consequently increase membrane fluidity and inhibit membrane-embedded enzymes [19]. The quality of dill seeds is related to their high content of carvone [20], with a minimum level of 30% in DSEO [21].

The composition of medical and culinary herbs is influenced by various environmental factors, primarily light quantity. Shade nets serve as a physiological tool to modify the crop microenvironment, enhancing plant growth, yield and phytonutrient composition [22]. Shading conditions have been observed to improve marjoram and oregano production [23,24,25] and positively impact the biological and chemical activity of culinary and medicinal plants, notably *Melissa officinalis* L. [26]. Shading has been shown to contribute to higher content of essential oils in *Melissa officinalis* L., *Mentha piperitae* L. and *Ocimum basilicum* L., with improved antioxidant and microbial activities [27]. The highest content of bioactive compounds was found in basil covered with red net [28]. Higher eugenol content and stronger antimicrobial activity were found in basil plants covered with blue nets [29]. The collection and study of different populations with morphological differences and molecular analyses can help in the selection of populations and varieties of culinary and medicinal plants [30].

The aim of the present study was to observe the effect of modified light conditions on the content, chemical composition and antioxidant activity of dill seed essential oil.

## 2. Materials and Methods

### 2.1. Plant Material and Growing Conditions

The experiment with local cultivar dill (cv. ‘Domaća aromatična’) was carried out in an experimental garden in the village of Moravac in south Serbia (21°42′ E, 43°30′ N, alt. 159 m a.s.l.) in 2022. 

The results of agrochemical analysis of the alluvial soil type in Table 1 show a fairly high humus content of 3.59% and a very good supply of phosphorus and potassium (40 mg/100 g). After deep plowing in the autumn and pre-sowing soil preparation in the spring, dill seed was sown in strips on a flat surface. 

The seeds were sown directly in the field at a rate of 5–7 kg seed per ha. The distance between the rows in the strip was 15–20 cm, and 50 cm between the strips, at a depth of 1.5–2 cm. Sowing took place on 20 May. After germination (1–3 weeks), the crop was thinned, so that a plant density of 50 plants/m^2^ could be achieved. Adequate water supply and optimal nutrition with an application of 80 kg N, 30 kg P and 30 kg K per ha were applied to maximize the vegetative mass yield, seed production and essential oil content. Flowering started about 60–90 days after sowing. 

Dill plants were covered with pearl nets with a shade index of 40% (Polysack, Nir Yitzhak, Israel), mounted on a structure placed about 2.0 m above the plants (net house) from mid-June until the end of August. The shading treatments along with un-netted control treatment were replicated three times (each with 50 plants) in a split-plot design.

Inter-row cultivation was implemented to protect dill plants against weeds, and a drip irrigation system was utilized. The optimal time for harvesting dill fell between 45 and 70 days after flowering, in early September, before the seeds began to fall. After separation from the umbel, the seeds were dried and stored in a dry and dark place until the moment of EO isolation (one month after harvest).

### 2.2. Clevenger Hydrodistillation

Disintegrated and homogenized dill seeds were used for EO isolation using Clevenger-type hydrodistillation, with the hydromodulus (the ratio of plant material:water) at 1:10 m/V for 120 min. The details with extended explanation are given in Stanojevic et al. [13]. The chemical composition of essential oil was studied in three replicates.

### 2.3. Gas Chromatography–Mass Spectrometry (GC/MS) and Gas Chromatography–Flame Ionization Detection (GC/FID) Analyses

The details of the method used are given in Milenković et al. [26] and Ilić et al. [27].

### 2.4. Antioxidant Activity—DPPH Assay

The ability of the EO to scavenge free DPPH radicals was determined using the DPPH assay. The details of the method used are given in Stanojevic et al. [28]. 

### 2.5. Antimicrobial Activity

Seven micro-organisms, six bacterial strains and one fungal strain were selected to determine the antimicrobial activity of essential oil: *Escherichia coli* (ATCC 25922), *Pseudomonas aeruginosa* (ATCC 27853), *Proteus vulgaris* (ATCC 8427), *Bacillus subtilis* (ATCC 6633), *Staphylococcus aureus* (ATCC 25923), *Klebsiella pneumoniae* (ATCC 700603) and (fungal strain) *Candida albicans* (ATCC 2091). Microorganisms were taken from the collection of the Microbiology Laboratory, Faculty of Technology, Leskovac. 

*Disc Diffusion Method.* The agar disc diffusion method was used to test the antimicrobial activity of the obtained extracts [31]. Bacterial and fungal suspensions were prepared using the direct colony method, as explained in Ilic et al. [27]. Inoculation and incubation were performed using the methods described in Stanojević et al. [13].

### 2.6. Statistical Analysis

Differences between the means were determined using one- and two-way ANOVA. TIBCO Software Inc. (Palo Alto, CA, USA). Data Science Workbench version 14 was used.

## 3. Results and Discussion

### 3.1. Climatic Conditions

Photoselective shade nets provide essential protection for vegetable plants cultivated in an open field or in a protected area from excessive solar radiation and high temperatures during the summer months. The importance of shading nets decreases during cloudy days. Therefore, common practice involves mounting the nets horizontally above the plants, enabling mobility for easy adjustment according to weather conditions. Shading nets reduce the intensity of light but also change its quality to a different level, which also affects the change in other environmental conditions [22]. Under pearl nets, photosynthetically active radiation (PAR) and solar radiation were significantly lower (more than 40%) compared to the control (open field condition) (Table 2).

Crop shading induces changes in the microclimate but also in plant activity. These microclimate changes are related to CO_2_ exchange, assimilation, and thus, indirectly, to the growth and development of plants [32] and secondary metabolite biosynthesis [33]. 

### 3.2. Yield of Fresh Mass, Dry Mass, Seed and Essential Oil

The yield of fresh biomass in shaded dill was 13.6 kg ha^−1^, which was statistically significantly higher (factor significance < 0.05) than in the control variant (10.8 kg ha^−1^). Similar results were reported in a study by Popović et al. [34]. Large dill biomass (stalks and umbels) can be used as feed additives and thus reduce the amount of waste. Higher yields (15.2–22.5 t/ha) of fresh mass compared to our results were found in the research of Wall and Friesen [21], while lower yields, depending on the cultivar (0.85–10.28 t ha^−1^), were reported by Bowes et al. [35].

In the shaded variant, the content of essential oil was 4.8%, which was higher (statistically significantly *p* < 0.05) than in the control variant (4.6%). Shaded plants showed a significantly higher average yield of essential oil (80.64 kg ha^−1^) in comparison with the yield from non-shaded plants (66.7 kg ha^−1^) (Table 3).

No differences were found in the number of umbels per plant between shaded and non-shaded plants (Table 3).

In a study on dill seed production, Wander and Bouwmeester [36] reported yields from 1.49 to 1.66 t ha^−1^, while in the research of Bailer et al. [37], seed yields from 0.80 to 1.5 t ha^−1^ were reported. The relatively lower seed yields were attributed to losses due to shattering [37].

Shading had no effect on umbel diameter, the number of umbels per plant or the ratio of external/internal parts of the umbel. Differences were statistically significant for seed weight per umbel and seed absolute mass. These parameters of yield were significantly different between the positions of umbels in plants. Thus, primary umbels in the plants had a statistically higher value with respect to secondary and tertiary umbels (Table 4). 

The interaction between shading and umbel position did not yield significant differences for all parameters. The content of EO depended on the plant part, origin, variety, the agricultural techniques applied, seasonal variations and the method of isolation. The EO content varied among different parts (leaves, flowers and seeds) of the same plant. The EO content in dill umbels was higher (2.0%) than in leaves (0.3%) [38], while seeds were richer in EO (3.4%) than flowers (3.2%) or umbels (1.2%) [39]. Regardless of the method of production, the content of DSEOs in our research (4.6–4.8%) was significantly higher than DSEO yields in previous research works from several different countries: Uzbekistan (4.2%) [39], Romania (3.4%) [38], Republic of Moldova (2.5–4%) [40], Egypt (3.2%) [16], Algeria (2.1%) [41] and Iran (1.2%) [42].

### 3.3. Dill Seed Essential Oil (DSEO) Compositions

The differences in the main constituents of DEO depend on the plant origin, growing conditions and method of production (nutrient, irrigation, sowing time, plant part), as well as the method and time of harvest and extraction methods [43,44]. The chemical composition of the EO obtained via hydrodistillation from seeds of dill was analyzed using GC/MS and GC/FID performed in triplicate to verify the repeatability of the analysis. Twenty-one DSEO compounds from non-shaded plants and twenty-two DSEO components from shaded plants were detected. The chromatogram (GC/FID) of dill seed essential oil from non-shaded dill plants is presented in Figure 1. 

The terpenic fraction of DSEO included oxygen-containing monoterpene derivatives, which constituted 53.6%, with the main components comprising carvone (46.1%) and trans-dihydrocarvone (6.8%), and monoterpene hydrocarbons (46.4%), which mainly included limonene (43.8%). The phenylpropanoids (found in traces, <0.05%), represented by (*E*)-anethole, were the least abundant (Table 5). 

DSEO isolated from the seeds of shaded dill plants (Table 5) contained a higher content of carvone (49.8%) and a lower content of limonene (37.8%) compared to seeds originating from non-shaded plants (46.1% and 43.8%, respectively). Monoterpene hydrocarbons (40.6%) were mainly represented by limonene (37.8%) and α-phellandrene (1.2%). Oxygenated monoterpenes (59.3%) were composed of carvone (49.8%), trans-dihydrocarvone (8.4%) and cis-dihydrocarvone (1.0%).

The chromatogram (GC/FID) of dill seed essential oil (DSEO) from non-shaded and shaded dill plants is presented in Figure 1.

The EOs, which were most present in dill seeds, are shown in Figure 2.

Comparing our results with earlier published data on DSEO composition revealed some similarities, as well as some differences. The quality of DSEO is determined by its carvone content. 

The results of the main components of DSEO originating from different countries are presented in Table 6. The percentage of carvone as the main dill seed EO constituent varied across countries as a result of genetic and/or environmental factors, which affect the biosynthesis of essential oils. It has been reported that the highest content of carvone was detected in DSEO from Romania (75.2%), Uzbekistan (73.61%) and Egypt (62.48%), while the lowest content was detected in DSEO from Algeria (34.3%).

Limonene was the dominant compound in Bulgarian (43.7%) samples [48], while dill from Saudi Arabia contained apiol and limonene as the dominant components [50]. α-phellandrene is the main component of DEO, with higher content in the leaf (62.71%) compared to the flower (30.2%) [39]. As dill plants mature, the content of dill EOs in the dill green mass decreases, while at the same time, the carvone content increases.

Through the stimulation of photosensitive enzymes (terpene synthases) involved in the mevalonic acid pathway, light intensity can affect essential oil production. Thus, light can directly influence the biosynthesis of aromatic compounds, or it can influence it indirectly, through the increase in plant biomass [51]. Parameters such as the cultivar, seeding date, harvesting stage, irrigation dose and plant density affect the volatile composition and dill essential oil components [52].

The specific novelty of the present study comprised optimized production techniques involving plants covered with shade nets, which could provide useful methods for improving the content, yield and composition of dill essential oils. For practical purposes, local domestic varieties of dill can serve as a good source of carvone and limonene, and the use of shade nets, especially under light-intense conditions, can increase the biosynthesis of EOs and their content.

### 3.4. Antioxidant Activity 

Based on the results of our experiment (Figure 3), it can be concluded that the antioxidant activity of DSEO in the previous results from Stanojević et al. [13], with an EC_50_ value of 20.27 mg mL^−1^, or in the research of Jianu et al. [53], with an EC_50_ value of 2.62 mg mL^−1^, had a stronger antioxidant effect on DPPH radical neutralization than in our research (EC_50_ = 26.04–54.23 mg mL^−1^). In a previous report, the DSEO extracted from Iranian dill plants showed a strong antioxidant activity, which was superior to the positive control (BHT) [54].

On the contrary, the DSEO from Thailand showed a lower extent of DPPH neutralization (EC_50_ = 128.49 mg mL^−1^) [55] compared to the EO obtained in our experiment. Carvone and its derivatives are powerful scavengers of free radicals due to the presence of an unsaturated hydroxyl group. DSEOs, as natural antioxidants, are an important source of alternative solutions in natural therapy. 

### 3.5. Antimicrobial Activity 

The antimicrobial activity of DSEO depends on the cultivation method, and the best antimicrobial activity was observed against *Escherichia coli* (inhibition zone of 15–18 mm in diameter) with a concentration of 10 μL. Furthermore, the antimicrobial activity of DSEO was significantly higher in non-shaded plants. Shading of dill plants showed no increased effect on the antimicrobial activity of DSEO against *Bacillus subtilis* and *Proteus vulgaris*. Shading positively influenced the antimicrobial activity of DSEOs against *Klebsiella pneumonia*, while no antimicrobial activity was observed against *Staphylococcus aureus* and *Candida albicans* (Table 7). 

The high antimicrobial activity of DSEOs against *E. coli* probably mainly originates from carvone (46.1–49.1%). This result is in accordance with studies by Stanojević and co-workers [13], who determined the effect of carvone. The antimicrobial activity of DSEO in our study likely originates from carvone, as suggested by published data on the antimicrobial effect of carvone on *Staphylococcus aureus*, *Bacillus subtilis* and *Escherichia coli* [13]. Previous research has shown that DSEO also showed significant antibacterial activities against *Staphylococcus aureus* and *Escherichia coli* [49]. Similarly, published data exist on limonene antimicrobial activity—a component represented at about 37.8–43.8% in isolated DSEO in this study. In similar studies, DSEOs rich in carvone and limonene exhibited high antimicrobial activity [56].

Dill oil showed significant-to-moderate antibacterial activity (10.0–15.0 mm inhibition zone) [57]. Future perspectives on this and similar studies have great potential for utilizing DEOs as alternatives and supplements to conventional antimicrobial additives in the food industry (against *Escherichia coli, Bacilus subtilis, Staphylococcus aureus* and *Candida albicans*). EOs, as volatile plant secondary metabolites, are used in medicine, cosmetics and the food industry due to their antimicrobial, antifungal and antioxidant activity [58]. Improvement in the antimicrobial efficiency of the EO can be promoted via nanoencapsulation due to the controlled release of EO aroma in food and protection from interactions with the environment (moisture, pH) [59]. The perspectives of using the encapsulation of clove essential oil (CEO) with chitosan emulsion/ionic gelation technique could improve the antimicrobial efficacy of EOs [60]. Therefore, CEO chitosan could be an interesting natural fungicide in agricultural and food technology. In the same way, EOs from other plant species loaded with chitosan could be included and would be suitable for the food and pharmaceutical industries due to their antibacterial activity (*E. coli* and *S. aureus*) and physicochemical properties [58].

Based on this research, plant cultivation under light modification can optimize dill seed production in order to obtain higher EO content with specific compounds. DSEO mainly exhibits antioxidant and antimicrobial potential and presents a wide array of applications in the food industry and storage of fresh fruits and vegetables.

## 4. Conclusions

Total seed yields ranged from 1.6 t ha^−1^ under non-shading conditions to 2.1 t ha^−1^ under shading conditions. The DSEO content varied from 4.63% in non-shaded plants to 4.81% in plants covered with shade nets. Significant DSEO yields were higher under shading (80.6 kg ha^−1^) conditions than under non-shading (66.7 kg ha^−1^) conditions. DSEO from shaded plants contained more carvone (49.8%) than limonene (37.8%) compared to DSEO from non-shaded plants (46.1% and 43.8%, respectively). Shade nets are recommended for achieving higher DSEO and higher carvone content but not higher antioxidant activity. DSEO from non-shaded plants had stronger antioxidant activity (26.04 mg mL^−1^) than DSEO from shaded plants (54.23 mg mL^−1^). DSEO showed the best antimicrobial activity against *Escherichia coli*. Shaded plants exhibited a positive influence of DSEO only against *Klebsiella pneumonia*. Future prospects should be geared in the direction of replacing synthetic oils with DSEO, which would reduce health and environmental problems.

## Figures and Tables

**Figure 1 plants-13-00886-f001:**
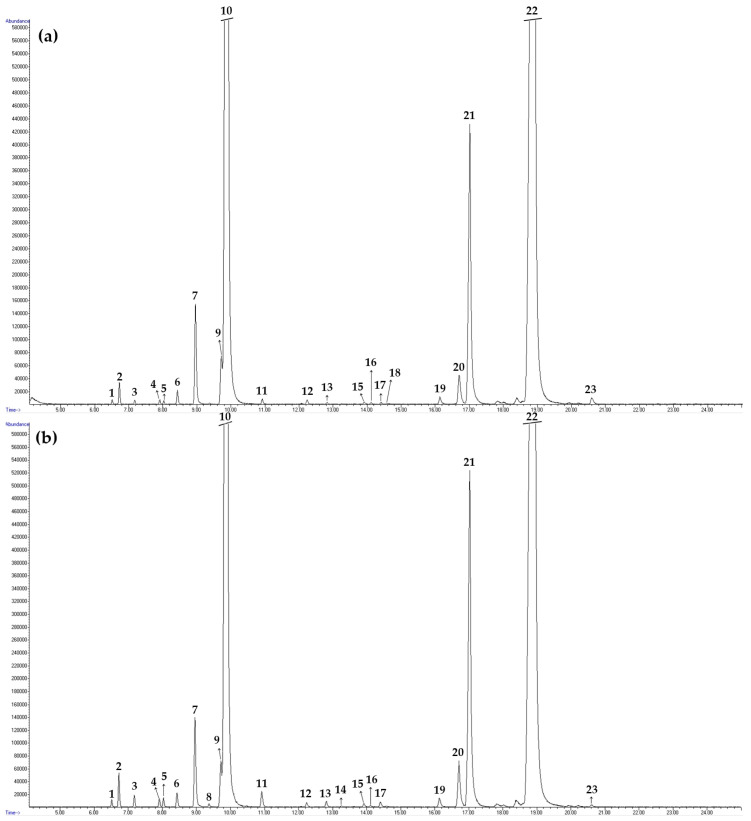
GC/MS chromatogram of essential oil isolated from non-shaded (**a**) and shaded (**b**) dill seeds.

**Figure 2 plants-13-00886-f002:**
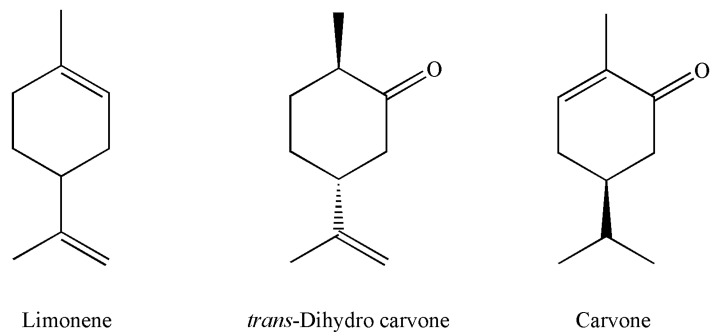
Chemical structures of the most abundant components of essential oil isolated from dill seeds.

**Figure 3 plants-13-00886-f003:**
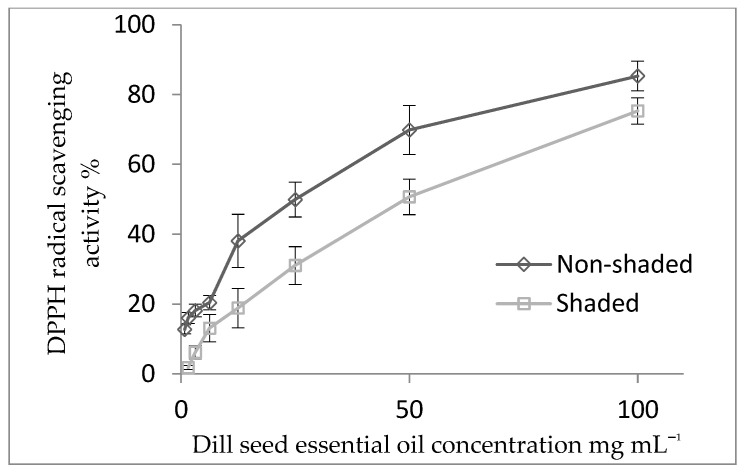
Antioxidant activity of dill seed essential oil (DSEO).

**Table 1 plants-13-00886-t001:** Chemical analysis of soil (0–30 cm in depth).

pH in 1 M KCl	pH in H_2_O	CaCO_3_ %	Humus %	N %	P_2_O_5_	K_2_O
					mg/100 g	
6.62	7.54	0.69	3.59	0.18	40	40

**Table 2 plants-13-00886-t002:** Influence of shading on growing environment (average day in July).

Time (h)	PAR * (μmol m^−2^ s^−1^)		Solar Radiation (W m^−2^)		Temperature °C		Relative Humidity %	
	Non- Shading	Shading Reduction %	Non- Shading	Shading	Non-Shading	Shading Reduction %	Non-Shading	Shading Reduction %
6:00	201.3	41.3	143.9	62.1	19.7	0.0	61.9	+0.6
9:00	1327.0	37.9	528.0	302.9	29.7	+2.15	40.8	+1.5
12:00	2106.5	40.3	884.5	482.1	36.6	+0.8	26.0	+3.1
15:00	1768.0	40.2	764.1	411.7	40.2	+0.17	18.9	−1.6
18:00	502.0	46.9	329.7	116.4	38.8	+0.51	28.2	0.0

* PAR—Photosynthetically active radiation.

**Table 3 plants-13-00886-t003:** Effect of shading on dill yield and parameters of yield.

Method of Production	Yield of Fresh Biomass kg/ha	Yield of Dry Mass kg/ha	Number of Umbels/Plants	Number of Non-Fertile Umbels	Seed Yield kg/ha	Content % EO in Seed	Yield of EO kg/ha
Control (non-shaded)	10.8 ^a^	2.73 ^a^	6.47 ^a^	1.9 ^b^	1.6 ^a^	4.6 ^a^	66.7 ^a^
Shaded plants	13.6 ^b^	3.40 ^b^	6.92 ^a^	1.0 ^a^	2.1 ^b^	4.8 ^b^	80.6 ^b^
ANOVA	*	*	NS	**	*	*	**

Means in columns marked with same letter are not statistically different. ANOVA significance: * *p* < 0.05; ** *p* < 0.01, NS: not significant.

**Table 4 plants-13-00886-t004:** The effect of shading and umbel position on the characteristics of dill seeds.

Method	Umbel	Umbel Diameter (cm)	Number of Umbelets/Umbel	Ratio of External/Internal Parts of Umbel	Seed Weight/Umbel (g)	Absolute Mass 1000 Seeds (g)
Non-shading	Primary	18.5 ^d^	20.7 ^c^	1.68 ^b^	1.7 ^d^	1.6 ^bc^
	Secondary	10.8 ^bc^	13.6 ^ab^	1.33 ^a^	0.8 ^c^	1.2 ^a^
	Tertiary	6.9 ^a^	10.8 ^a^	-	0.2 ^a^	1.2 ^a^
Shading	Primary	18.2 ^d^	20.7 ^c^	1.62 ^ab^	2.1 ^e^	1.9 ^d^
	Secondary	12.8 ^c^	16.8 ^b^	1.44 ^ab^	0.8 ^c^	1.7 ^cd^
	Tertiary	9.4 ^ab^	13.2 ^ab^	-	0.5 ^b^	1.4 ^ab^
	Shading	NS	NS	NS	**	**
	Umbel	**	**	*	**	**
	Shading x Umbel	NS	NS	NS	NS	NS

Means in columns marked with same letter are not statistically different. FACTOR and interaction significance: * *p* < 0.05; ** *p* < 0.01, NS: not significant.

**Table 5 plants-13-00886-t005:** Chemical composition of dill seed essential oil (DSEO) from non-shaded and shaded plants.

N^o^	*t*_ret_, min	Compound	RI^exp^	RI^lit^	Method of Identification	Content %	
						Non-Shaded	Shaded Plants
1.	6.51	α-Thujene	927	924	RI, MS	tr	tr
2.	6.73	α-Pinene	934	932	RI, MS, Co-I	0.2 ± 0.01	0.3 ± 0.01
3.	7.18	Camphene	950	946	RI, MS	tr	0.1 ± 0.00
4.	7.91	Sabinene	975	969	RI, MS	tr	tr
5.	8.03	β-Pinene	979	974	RI, MS	tr	tr
6.	8.43	Myrcene	993	988	RI, MS	0.2 ± 0.01	0.2 ± 0.00
7.	8.96	α-Phellandrene	1008	1002	RI, MS	1.4 ± 0.02	1.2 ± 0.02
8.	9.37	α-Terpinene	1020	1014	RI, MS	-	tr
9.	9.71	*p*-Cymene	1028	1020	RI, MS	0.7 ± 0.01	0.7 ± 0.01
10.	9.91	Limonene	1032	1024	RI, MS, Co-I	43.8 ± 0.05	37.8 ± 0.43
11.	10.93	γ-Terpinene	1061	1054	RI, MS	tr	0.2 ± 0.00
12.	12.24	*p*-Cymenene	1097	1089	RI, MS	tr	tr
13.	12.82	*cis*-Thujone	1111	1101	RI, MS	tr	tr
14.	13.25	*trans*-Thujone	1121	1112	RI, MS	-	tr
15.	13.92	*cis*-Limonene oxide	1137	1132	RI, MS	tr	tr
16.	14.12	*trans*-Limonene oxide	1142	1137	RI, MS	tr	tr
17.	14.41	Camphor	1149	1141	RI, MS, Co-I	tr	tr
18.	14.58	Myrcenone	1153	1145	RI, MS	tr	-
19.	16.13	Dill ether	1190	1184	RI, MS	tr	0.2 ± 0.00
20.	16.70	*cis*-Dihydrocarvone	1201	1191	RI, MS	0.7 ± 0.01	1.0 ± 0.02
21.	17.02	*trans*-Dihydrocarvone	1210	1200	RI, MS	6.8 ± 0.08	8.4 ± 0.13
22.	18.91	Carvone	1249	1239	RI, MS	46.1 ± 0.06	49.8 ± 0.62
23.	20.60	(*E*)-Anethole	1292	1282	RI, MS, Co-I	tr	tr
				Total identified (%)		100.0	100.0
	Grouped components (%)						
	Monoterpene hydrocarbons (**1**–**12**)					46.4 ± 0.06	40.6 ± 0.47
	Oxygen-containing monoterpenes (**13**–**18**, **20**–**22**)					53.6 ± 0.15	59.3 ± 0.77
	Phenylpropanoids (**23**)					tr	tr
	Others (**19**)					tr	0.2 ± 0.00

*t*_ret_—retention time; RI^lit^—retention indices from the literature (Adams, 2007) [45]. RI^exp^—experimentally determined retention indices using a homologous series of *n*-alkanes (C_8_–C_20_) on the HP-5MS column; MS—constituent identified by mass-spectra comparison; RI—constituent identified by retention index matching; Co-I—constituent identity confirmed by GC co-injection of an authentic sample; tr—trace amount (<0.05%).

**Table 6 plants-13-00886-t006:** Main components of DSEO in different countries.

Dill Origin (Country)	Main Components of Dill Seed	Method of Isolation	Reference
Egypt	Carvone (62.48%), dillapiole (19.51%) and limonene (14.61%)	Liquid chromatography (GLC) analysis	Said-Al Ahl et al., 2015 [4]
Tajikistan	Carvone (51.7%), trans-dihydrocarvone (14.7%), dill ether (13.2%), α-phellandrene (8.1%) and limonene (6.9%)	GC/MS using an Agilent 6890 GC with Agilent 5973 mass selective detector	Sharopov et al., 2013 [46]
Iran	Carvone (57.3%) and limonene (33.2%)	GC and GC/MS	Sefidkon, 2001 [47]
Bulgaria	Limonene (43.7%), carvone (41.2%), dihydrocarvone (3.1%) and myristicin (11.70%)	GC and GC/MS	Kruger and Hammer [48]
Algeria	Carvone (34.33%), α-phellandrene (22.03%) and dill ether (18.84%)	GC and GC/MS	Benlembarek et al., 2022 [49]
Romania	Carvone (75.2%) and limonene (21.56%)	GC/MS analyses	Rădulescu et al., 2010 [38]
Saudi Arabia	Apiol (33.3%), limonene (30.8%) and carvone (17.70%)	GC/MS analyses	Aati et al., 2022 [11]
India	Carvone (41.15%), limonene (23.11%) and camphor (9.25%)	GC/MS analyses	Chahal et al., 2016 [50]
Uzbekistan	Carvone (73.61%), limonene (14.69%) and cis-dihydrocarvone (5.87%)	GC/MS	Yili et al., 2016 [39]

**Table 7 plants-13-00886-t007:** Antimicrobial activity of dill seed essential oil (DSEO).

Method of Plant Production	*Escherichia* *coli*	*Proteus* *vulgaris*	*Bacillus* *subtilis*	*Staphylococcus* *aureus*	*Klebsiella* *pneumoniae*	*Candida* *albicans*
	Inhibition Zone (mm)					
Non-shaded plants	18.0 ^b^	11.3 ^b^	12.3 ^b^	n.z.	11.0 ^c^	n.z.
Shaded plants	15.0 ^c^	11.5 ^b^	12.0 ^b^	n.z.	13.0 ^b^	n.z.
Positive control (Ceftriaxone 30 μg for bacteria and Nystatin 50 μg for yeast)	32.0 ^a^	30.0 ^a^	24.0 ^a^	25.0	20.0 ^a^	17.0
Shading	***	*	*	-	***	-

n.z.—no zone; Means in columns marked with same letter are not statistically different; *—no statistically significant differences; ***—differences are significant.

## Data Availability

The data presented in this study are publicly available.

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
