# Peer review of "Chemical Composition and Bioactivity of Dill Seed (Anethum graveolens L.) Essential Oil from Plants Grown under Shading"

_plants, 2024, doi:10.3390/plants13060886_

Round 1
Reviewer 1 Report
Comments and Suggestions for Authors
Manuscript (plants-2893582) entitled ‘Chemical Composition and Bioactivity of Dill Seed (Anethum graveolens L.) Essential Oil from Plants Grown under Shading’ by Zoran Ilic, Lidija Milenković, Ljiljana Stanojevic, Bojana Danilovic, Ljubomir Sunic, Zarko Kevresan, Jelena Stanojevic and Dragan Cvetkovic presents interesting results about essential oil extracted from shade and non-shaded dill seeds. Introduction is focused on the paper, methodologies are given in detail and discussion needs some elaboration. English language needs checking for errors. Authors usually write statistical instead statistically, should be changed.
Authors may find below additional comments and remarks to improve manuscript.
Abstract:
Line 18: were
I suggest to accompany ‘antioxidant activity’ with EC 50
I suggest to accompany ‘antimicrobial activity’ with disc diffusion method
Introduction:
I suggest to very shortly describe antimicrobial activity of dill extracts
Material and methods
Gas chromatography – mass spectrometry should be new subchapter not in 2.3
For all parameters studied, units how the results are expressed should be given
Results and discussion
Line 247: The phenylpropanoids (tr) – what is tr it should more precisely explained
Table 5: I suggest to add statistics
Conclusions: I suggest to add future prospects for study like this.
Comments on the Quality of English LanguageEnglish language needs checking for errors.
Author Response
Authors may find below additional comments and remarks to improve manuscript.
Authors usually write statistical instead statistically, should be changed.
Abstract:
Line 18: were
I suggest to accompany ‘antioxidant activity’ with EC 50
I suggest to accompany ‘antimicrobial activity’ with disc diffusion method
We adopt your recommendation
Introduction:
I suggest to very shortly describe antimicrobial activity of dill extracts
Monoterpenes (limonene and carvone) displaying potent antibacterial activity. Monoterpenes have lipophilic characters, allowing them to permeate into membrane structure and consequently increase membrane fluidity and inhibit membrane -embedded enzymes.
Material and methods
Gas chromatography – mass spectrometry should be new subchapter not in 2.3
We create new subtitle
2.3. Gas chromatography-mass spectrometry (GC/MS) and gas chromatography-flame ionization detection (GC/FID) analysis.
The details of the method used are given in Milenković et al.2021 and Ilić et al [27,28].
2.4. Antioxidant activity. DPPH assay
The ability of the EO to scavenge free DPPH radicals was determined using the DPPH assay. The details of the method used are given in Stanojevic et al. [13].
For all parameters studied, units how the results are expressed should be given
All unit were present in the text body
Results and discussion
Line 247: The phenylpropanoids (tr) – what is tr it should more precisely explained
We are excude this sentence
Table 5: I suggest to add statistics
We add standard deviations in Table 5
Conclusions: I suggest to add future prospects for study like this
We add new paragraph
For practical purposes, local domestic varieties of dill can serve as a good source of carvone and limonene, and the use shade nets, especially in high light conditions, can increase biosynthesis of EO and their content.
Future prospects should go in the direction of replacing synthetic oils with DSEO, which would reduce health and environmental problems.
Reviewer 2 Report
Comments and Suggestions for Authors
The paper is well structured, the manuscript meets with the author’s guidelines and Aims & Scopes of the Journal Plants. The similarity is of 24% as detected by the Compilatio software. The similarity and plagiarism detection is mostly related to the mention of 12/65 references in autocitation (18%).
Although the methodology brings out the effect of light Dill seed’s composition, the biological activity: antioxidant and antimicrobial have been previously published in 2016. The reference is mentioned for the antioxidant activity analysis but is lacking for the antimicrobial activity.
The extraction methods have been previously reported. The results show the activity of essential oil extracts. The GC-MS results showing the major components is not consistent with the authors description in page 7 L. 276-277.
The purification and identification of the major components could bring insight to the potential application in cosmetic industry and the complementary toxicology analysis.
The figures could be provided with a higher resolution (Figures 1, 2 and 3).
I recommend this paper to be published in the Journal Plants, but there are some specific (major revision) to achieve the purification, isolation, and analysis of the mentioned components separately) as well as adding references/control comparatively to the EO’s antimicrobial activity. Further toxicologic evaluation would present the perspectives of using the molecules in nanoencapsulation as mentioned by the authors (P.7 L. 357).
Author Response
Reviewer 2
The “2. Material and methods, 2.1. Plant material and growing conditions” should only contain a description of the plant material and growing condition. The results and the discussion should be given elsewhere.
We are adopting your recommendation!!
In “2.3. Antioxidant activity. DPPH assay” the sentence “Gas chromatography-mass spectrometry (GC/MS) and gas chromatography-flame ionization detection (GC/FID) analysis” is missing verb.
We created new subparagraph 2.3 and 2.4
Under Table 3 two rows begin with “LSD” whose meaning is unclear to me. Needs some explanation.
We make new statistical analysis one way Anova …..for Table 3
The same is true with Table 4. What is the meaning of the “AxB” row?
We make new statistical methods two way ANOVA for Table 4
How many samples were used to determine the amount of the essential oils? What is the standard deviation?
We add standard deviation
Standard deviations are missing in Tables 5 and 6.
We add standard deviations in Table 5
It would be better to combine Figures 1 and 2 into one.
Yes we adopt your suggestion for Figure 1 and 2. . We created new one
In Figure 4: the error bars are missing.
We add error bars in Figure 4

Reviewer 3 Report
Comments and Suggestions for Authors
The “2. Material and methods, 2.1. Plant material and growing conditions” should only contain a description of the plant material and growing condition. The results and the discussion should be given elsewhere.
In “2.3. Antioxidant activity. DPPH assay” the sentence “Gas chromatography-mass spectrometry (GC/MS) and gas chromatography-flame ionization detection (GC/FID) analysis” is missing verb.
Under Table 3 two rows begin with “LSD” whose meaning is unclear to me. Needs some explanation. The same is true with Table 4. What is the meaning of the “AxB” row?
How many samples were used to determine the amount of the essential oils? What is the standard deviation?
Standard deviations are missing in Tables 5 and 6.
It would be better to combine Figures 1 and 2 into one.
In Figure 4: the error bars are missing.
This research missing any statistics.
Author Response
Reviewer 3
The paper is well structured, the manuscript meets with the author’s guidelines and Aims & Scopes of the Journal Plants. The similarity is of 24% as detected by the Compilatio software. The similarity and plagiarism detection is mostly related to the mention of 12/65 references in autocitation (18%).
Although the methodology brings out the effect of light Dill seed’s composition, the biological activity: antioxidant and antimicrobial have been previously published in 2016. The reference is mentioned for the antioxidant activity analysis but is lacking for the antimicrobial activity.
The extraction methods have been previously reported. The results show the activity of essential oil extracts. The GC-MS results showing the major components is not consistent with the authors description in page 7 L. 276-277.
We have made changes
The purification and identification of the major components could bring insight to the potential application in cosmetic industry and the complementary toxicology analysis.
The figures could be provided with a higher resolution (Figures 1, 2 and 3).
We provided Figures with higher resolution
I recommend this paper to be published in the Journal Plants, but there are some specific (major revision) to achieve the purification, isolation, and analysis of the mentioned components separately) as well as adding references/control comparatively to the EO’s antimicrobial activity. Further toxicologic evaluation would present the perspectives of using the molecules in nanoencapsulation as mentioned by the authors (P.7 L. 357)
We have added a paragraph with additional information about EO’s antimicrobial activity.

Round 2
Reviewer 2 Report
Comments and Suggestions for Authors
The authors have made adjustments according to the reviewers suggestions.
L100 : "Secondary metabolites content it is related much more to the specificity of the species than to shading": the authors should remove "it".
L133-134: Revise the sentence: "The seeds easily separated from fruits (shizocarp), dried and placed in a dry and dark place until the moment when the composition of its EO will be analyzed" .
The authors should explain the number of replicas and samples.
the analysis of the most abundant compounds in shaded vs. non shaded is not analyzed and the discussion could bring insight into the light effect on secondary metabolites production.
Among the most abundant components Carvone is available at 46% in contradiction with the component 22 on the GC/MS.
The paragraph highlighting the difference in chemical composition comparatively to different countris should take into consideration the extraction method and could be better presented in a Table to bring a better understanding of the applied conditions.
The antibacterial analysis still lacks a positive control analysis.
The administration of Essential oils should be provided with cell viability tests.
Author Response
The authors have made adjustments according to the reviewers suggestions.
L100 : "Secondary metabolites content it is related much more to the specificity of the species than to shading": the authors should remove "it".
Yes, we are removing
L133-134: Revise the sentence: "The seeds easily separated from fruits (shizocarp), dried and placed in a dry and dark place until the moment when the composition of its EO will be analyzed" .
After separation from umbel, the seed are dried and placed in a dry and dark place until the moment of EO isolation (one month after harvest).
The authors should explain the number of replicas and samples.
Combinations of plant shading treatments and un-netted control treatment were replicated 3 times (with 50 plants) in a split-plot design.
All seed samples for EO isolation and analysis was done three times
the analysis of the most abundant compounds in shaded vs. non shaded is not analyzed and the discussion could bring insight into the light effect on secondary metabolites production.
Through the stimulation of photosensitive enzymes (terpene synthases) involved in the mevalonic acid pathway, light intensity can affect essential oil production. Thus, light can directly influence the biosynthesis of aromatic compounds or indirectly, through the increase of plant biomass [55]. Parameters such as, cultivar, seeding date, harvesting stage, irrigation dose and plant density affect the volatile composition and dill essential oil components [56].
Specific and novelty of the present study was optimized production techniques using plant cover by shade nets which could provide useful methods for improving the content, yield and composition of dill essential oils. For practical purposes, local domestic varieties of dill can serve as a good source of carvone and limonene, and the use shade nets, especially in high light conditions, can increase biosynthesis of EOs and their content.
Among the most abundant components Carvone is available at 46% in contradiction with the component 22 on the GC/MS.
Figure was improve
The paragraph highlighting the difference in chemical composition comparatively to different countris should take into consideration the extraction method and could be better presented in a Table to bring a better understanding of the applied conditions.
The results of the present in Table 6. showed various percent of carvone as main dill seed constituents of plants originate vary with different country. Genetic or environmental factors affect the biosynthesis of essential oils in a particular plant. It has been reported that highest content of carvone in DSEO from Romania (75.2%), Uzbekistan (73.61%) and Egypt (62.48%) or the lowest from Algeria (34.3%).
Table 6. Main components of DSEO from different country
|
Dill origin (country) |
Main components in dill seed |
Method of isolation |
Reference |
|
Egypt |
carvone (62.48%), dillapiole (19.51%) and limonene (14.61%) |
liquid chromatographic (GLC) analysis. |
Hussein et al., 2015 [48] |
|
Tajikistan |
carvone (51.7%), trans-dihydrocarvone (14.7%), dill ether (13.2%), α-phellandrene (8.1%) and limonene (6.9%). |
GC-MS using an Agilent 6890 GC with Agilent 5973 mass selective detector
|
Sharopov et al., 2013 [49] |
|
Iran |
carvone (57.3%), limonene (33.2%) |
GC and GC/MS |
Sefidkon, 2001 [50] |
|
Bulgaria |
limonene (43.7%), carvone (41.2%), dihydrocarvone (3.1%), myristicin (11.70%) |
GC and GC/MS |
Kruger & Hammer [51] |
|
Algeria |
carvone (34.33%), α-phellandrene (22.03%), dill ether (18.84%) |
GC and GC/MS. |
Benlembarek et al. 2022 [52] |
|
Romania |
carvone 75.2%, limonene 21.56% |
GC-MS analyses |
Rădulescu et al., 2010 [40] |
|
Saudi Arabian |
apiol (33.3%), limonene (30.8%), and carvone (17.70%) |
GC-MS analyses |
Aati et al., 2022 [53] |
|
India
Uzbekistan |
Carvone (41.15%), limonene (23.11%), camphor (9.25%)
carvone (73.61%), limonene (14.69%), cis-dihydrocarvone (5.87%) |
GC-MS analyses
GC- MS |
Chalal et al., 2016 [54]
Yili et al., 2016 [41] |
|
|
|
|
|
Limonene was the dominant compound in Bulgarian (43.7%) samples [51], while dill from Saudi Arabian contain apiol and limonene as dominant components [53]. α-phellandrene is the main component of DEO and is more highly represented in the leaf (62.71%) than in the flower (30.2) [41]. As dill plants mature the content of dill ether in the dill green mass EOs decreases, while carvone content increases.
The antibacterial analysis still lacks a positive control analysis.
We add positive Control
Table 7. Antimicrobial activity of dill seeds essential oil (DSEO)
|
Method of plant production |
Escherichia coli |
Proteus vulgaris |
Bacillus subtilis |
Staphylococcus aureus |
Klebsiella pneumoniae |
Candida albicans |
|
Inhibition zone (mm) |
||||||
|
Non-shaded plants |
18.0b |
11.3b |
12.3b |
n.z. |
11.0c |
n.z. |
|
Shaded plants |
15.0c |
11.5b |
12.0b |
n.z. |
13.0b |
n.z. |
|
Positive control (Ceftriaxone 30 μg for bacteria and Nystatin 50 μg for yeast) |
32.0a |
30.0a |
24.0a |
25.0 |
20.0a |
17.0 |
|
Shading |
*** |
* |
* |
- |
*** |
- |
n.z. - no zone
*no statistical significant differences; ***differences is significant
The administration of Essential oils should be provided with cell viability tests.
It will be the subject of our further research.
Reviewer 3 Report
Comments and Suggestions for Authors
The authors have complied with most of my comments. The manuscript is ready for publishing in its present form. However, checking for any typographical or grammatical errors would be of help.
Author Response
Reviewer are satisfied with last correction
Round 3
Reviewer 2 Report
Comments and Suggestions for Authors
The authors have replied to the Reviewers suggestions.
Author Response
Reviewer 2 is satisfied with last version, and havent new request !!!!